# In Vitro Fermentation of Edible Mushrooms: Effects on Faecal Microbiota Characteristics of Autistic and Neurotypical Children

**DOI:** 10.3390/microorganisms11020414

**Published:** 2023-02-06

**Authors:** Georgia Saxami, Evdokia K. Mitsou, Evangelia N. Kerezoudi, Ioanna Mavrouli, Marigoula Vlassopoulou, Georgios Koutrotsios, Konstantinos C. Mountzouris, Georgios I. Zervakis, Adamantini Kyriacou

**Affiliations:** 1Department of Nutrition and Dietetics, Harokopio University, 17671 Athens, Greece; 2School of Medical Sciences, Faculty of Medicine and Health, Örebro University, SE-701 82 Örebro, Sweden; 3Laboratory of General and Agricultural Microbiology, Department of Crop Science, Agricultural University of Athens, 11855 Athens, Greece; 4Department of Nutritional Physiology and Feeding, Agricultural University of Athens, 11855 Athens, Greece

**Keywords:** autism spectrum disorders, *Pleurotus eryngii*, *Pleurotus ostreatus*, in vitro fermentation, gut microbiota analysis, short-chain fatty acid production

## Abstract

Children with autism spectrum disorder (ASD) often suffer gastrointestinal disturbances consistent with gut microbiota (GM) alterations. Treatment with pro/prebiotics may potentially alleviate gut symptoms, but the evidence for prebiotics is scarce. This study aims to evaluate the effects of edible mushrooms (*Pleurotus*, Basidiomycota) and prebiotic compounds on GM composition and metabolite production in vitro, using faecal samples from autistic and non-autistic children. Specific microbial populations were enumerated after 24 h of fermentation by quantitative PCR, and the metabolic production was determined by gas chromatography. Higher levels of *Prevotella* spp. and *Bifidobacterium* spp. were measured in neurotypical children compared to ASD children. A total of 24 h fermentation of *Pleurotus eryngii* and *P. ostreatus* mushroom powder increased the levels of *Bifidobacterium*, while known prebiotics increased the levels of total bacteria and *Bacteroides* in both groups. Only *P. eryngii* mushrooms resulted in significantly elevated levels of total bacteria *Bacteroides* and *Feacalibacterium prausnitzii* compared to the negative control (NC) in the ASD group. Both mushrooms induced elevated levels of butyrate after 24 h of fermentation, while short-chain fructooligosaccharides induced increased levels of acetate in the ASD group, compared to NC. Overall, this study highlights the positive effect of edible mushrooms on the GM and metabolic activity of children with ASD.

## 1. Introduction

Autism spectrum disorder (ASD) is a group of neurodevelopmental disorders that are characterized by abnormal social interactions and communication, together with repetitive and stereotyped patterns of behavior and abnormal sensory responses [1]. Recent epidemiological studies reported that the prevalence of ASD was 1.70 and 1.85% in US children aged 4 and 8 years, respectively, while the prevalence in Europe ranged between 0.38 and 1.55% [2]. The etiology of ASD remains unknown. Various factors have been associated with the development of ASD, including both genetic and environmental factors [3]. Although the exact etiopathogenesis of ASD has not been elucidated, recent studies have highlighted the effect of the gut-brain axis in patients with autism or other neuropsychiatric diseases [4,5].

The gut–brain axis is defined as a bidirectional communication system comprised of the Central Nervous System (CNS), intrinsic branches of the enteric nervous system (ENS), extrinsic parasympathetic and sympathetic branches of the Autonomic Nervous System (ANS), the hypothalamic-pituitary-adrenal axis (HPA), neuroimmune pathways (neurotransmitters, hormones, and neuropeptides), and the intestinal microenvironment [6,7]. Children with ASD are more likely to experience gastrointestinal disturbances such as constipation, gaseousness, diarrhea, flatulence, increased intestinal permeability, and abdominal pain [8,9]. The gastrointestinal symptoms of individuals with ASD seem to be significantly correlated with the degree of behavioral and cognitive impairment. For example, in individuals with ASD, irritability, aggression, sleep disturbances, and self-injury are strongly associated with gastrointestinal (GI) symptoms [9,10]. This evidence suggests that gastrointestinal abnormalities, perhaps linked to gut dysbiosis, may be associated with ASD [7]. The gut microbiota (GM) is one of the most complex and dense microbial ecosystems, and is recognized for its crucial role in human health. Imbalance in the dynamic interactions among microbial intestinal populations, a phenomenon called dysbiosis, has been related to many pathological conditions, e.g., inflammatory bowel disease, irritable bowel syndrome, obesity, and colorectal cancer [11,12].

Various emerging studies have revealed alterations in the gut microbiota composition in ASD individuals compared to neurotypical children [13]. It should be noted that no specific microbial species has been found to differ substantially across ASD-related studies, as different factors such as diet, age, gender, population, and autism severity should have been considered [13]. Although alterations in GM in ASD patients are not always consistent across studies, ASD patients often exhibit microbial imbalances of several types, most notably a higher abundance of the genera *Bacteroides*, *Parabacteroides*, *Clostridium*, *Faecalibacterium*, and *Phascolarctobacterium*, and a lower relative abundance of *Streptococcus* and *Bifidobacterium* [9,14].

Metabolic alterations have also been identified in ASD and may be attributed to gut dysbiosis in autistic children. Short-chain fatty acids (SCFAs), mainly acetate (AA), propionate (PPA), and butyrate (BA), are by far the most investigated bacterial-produced metabolites involved in health and disease. Complex carbohydrates or dietary fibers that escape the host’s digestion are the main substrates for the production of SCFAs by gut microbiota. These bioactive compounds have been associated with many beneficial effects, such as the following: the inhibition of pH-sensitive pathogens; the increase in mineral absorption; the stimulation of gut motility and epithelial barrier integrity; and the effects on brain development and social behavior [15]. Although the data are slightly inconsistent between different studies, propionate has been found to be augmented in individuals with ASD, while butyrate has been shown to be significantly downregulated [16,17].

Improvement of gut symptoms in ASD with pro/prebiotics has been postulated, but the evidence for the effects of prebiotics on ASD is scarce [16]. As defined by the International Scientific Association for Probiotics and Prebiotics (ISAPP), prebiotics are substrates “selectively utilized by host microorganisms, conferring a health benefit” [18]. The most recognized prebiotics are fructo-oligosaccharides (FOS), galacto-oligosaccharides (GOS), lactulose, and inulin, whereas β-glucans derived from various mushroom species (i.e., *Pleurotus ostreatus* and *Pleurotus eryngii*) are potential prebiotic candidates [19]. Recently, whole food-based strategies (not only specific bioactive ingredients) have been applied in order to modulate GM via potential synergistic interactions among different components of the food [20]. Our group demonstrated that lyophilized powder from the entire fruitbodies of *Pleurotus* and *Cyclocybe* strains exhibited a beneficial effect on the GM composition of apparently healthy subjects over 65 years old, highlighting their potential as candidate prebiotics [21]. More recently, we illustrated the preventive effect of fermentation supernatants derived from *P. eryngii* mushrooms against imminent damage to the intestinal barrier through upregulation of tight junctions’ genes, fueling further efforts to clarify their manifold role, prebiotics included, in human health [19].

In order to examine the impact of prebiotic compounds on gut microbial composition and metabolic activity, in vitro gut fermentation models have been developed as a powerful tool [22]. Several studies have been conducted to investigate the prebiotic potential of faecal inoculum from infants or healthy adults, but there are only a few relevant studies for ASD. In addition, data from the in vitro fermentation of edible mushrooms (i.e., lyophilized powder from the entire fruitbody) using faecal samples derived from ASD individuals are missing. 

Our study aimed to assess the effects of edible mushrooms, as well as of known prebiotic compounds, on gut microbiota composition and metabolite production using an in vitro batch culture fermentation system inoculated with faecal samples of autistic and non-autistic children.

## 2. Materials and Methods

### 2.1. Fermentation Substrates 

Lyophilized mushroom powder was obtained after the cultivation of *Pleurotus eryngii* strain LGAM 216 and *Pleurotus ostreatus* strain IK 1123. Pure cultures of these strains are preserved in the fungal culture collection of the Laboratory of General and Agricultural Microbiology (Agricultural University of Athens, Athens, Greece). Cultivation substrates (based on wheat straw) and conditions for mushroom production have been previously described [23]. The documented prebiotic inulin (Orafti^®^ GR, BENEO-Orafti, Oreye, Belgium) and short-chain fructooligosaccharides (Actilight 950P, Beghin Meiji Merckolsheim France) were used as positive controls for the fermentation procedure. A negative control (NC) was also included, i.e., a basal medium with no carbohydrate source. The treatments used in this study are presented in Table 1.

### 2.2. Faecal Donors’ Characteristics

Faecal samples were obtained from five autistic child donors (male, aged 3.5–10 years old) and five age-matched typically developing donors (serving as controls) meeting the following inclusion criteria: (a) no history of gastrointestinal disease, chronic constipation, chronic/acute diarrhea, autoimmune disease, coronary disease, liver and/or kidney malfunction; (b) no antibiotic use two months prior to the study; and (c) no use of probiotics, prebiotics, and/or dietary fibre supplements two weeks prior to the study. All autistic children had a formal diagnosis of mild autism. Subjects completed a series of questionnaires in relation to sociodemographic parameters (including age and sex). Evacuation characteristics were also recorded for the past 7 days prior to faecal sampling. Body weight and height were measured by a dietician on a levelled platform scale (SECA GmbH, Hamburg, Germany) and a wall-mounted stadiometer (SECA GmbH) to the nearest 0.1 kg and 0.5 cm, respectively. Body mass index (BMI) was calculated by dividing the weight (kg) by the height (m^2^). Dietary intake was evaluated through a 3D food diary, and data were analyzed in terms of energy and nutrient intakes, using the Nutritionist Pro software (Version 4.1.0.; Axxya Systems, Staord, TX, USA).

### 2.3. Ethical Standards

All parents were provided with written informed consent for the use of their children’s faeces in the study. This study was conducted according to the guidelines laid down in the Declaration of Helsinki and with the approval of the Bioethics Committee of Harokopio University (7/18 June 2019). 

### 2.4. Fecal Sample Collection and In Vitro Static Batch Culture Fermentations

Parents were given a pre-weighed plastic container to collect and return their children’s entire evacuation over the next few days. Stool samples were weighted, homogenized, and processed within two hours of defecation. The in vitro static batch culture fermentation process was performed according to the protocols of Olano–Martin et al. [24] and Rycroft et al. [25] with slight modifications. The composition of the basal medium was further modified, as previously described [21]. The medium was pH-controlled at 7.0 with HCl 1.0 M. Volumes of 45 mL were aliquoted into 100 mL vessels, sterilised at 121 °C for 15 min, and transferred into the anaerobic chamber (BACTRON900 Anaerobic Chamber, Sheldon Manufacturing Inc., Cornelius, OR, USA) for a 12 h overnight pre-reduction the day before the in vitro static batch culture fermentation process.

On the day of the in vitro experiment, 2% (*w*/*v*) of the lyophilized mushrooms powder, or 2% (*w*/*v*) of the inulin, scFOS, or NC were added to the basal medium aliquots. A faecal slurry (20% *w*/*v*) was prepared in PBS pH 7.3 [26] and homogenized manually for approximately 20 s. From this slurry, 10% (*v*/*v*) inocula were transferred into pre-reduced basal medium aliquots containing the tested mushrooms or controls. The cultures were incubated for 24 h at 37 °C under anaerobic conditions with stirring. Samples were collected at 0 h, 8 h, and 24 h of fermentation and stored at −80 °C until further analysis.

### 2.5. Quantitative Polymerase Chain Reaction (qPCR) Enumeration of Gut Microbiota In Vitro

Bacterial composition of total bacterial load and selected members of the GM were enumerated at the beginning t = 0 h) and after 24 h of fermentation by real-time quantitative PCR (qPCR) as previously described [21]. For gut microbiota analysis, we have selected bacteria with previously reported alterations in autism spectrum disorders [16,27]. Quantitative polymerase chain reaction based on SYBR Green I was performed on a LightCycler^®^ 2.0 Real-Time PCR System (Roche Diagnostics GmbH, Mannheim, Germany) using the KAPA SYBR^®^ Fast Master Mix (2×) Universal Kit (Kapa Biosystems Inc., Wilmington, MA, USA) (Table 2) [21]. PCR reactions were performed in duplicate, in LightCycler^®^ glass capillaries, and contained 10 ng of each faecal DNA preparation (2 ng ΜL^−1^), 10 μL of KAPA kit, 200 nM of each primer, 0.25 μL of Bovine Serum Albumin (BSA 20 mg mL^−1^, New England Biolabs Inc, Hitchin, UK), and 3.95 μL ddH_2_O. The thermal cycling conditions consisted of 3 min at 95 °C, followed by 45 cycles of 3 s at 95 °C, and then 20 s at 72 °C. Primer specificity was verified by performing a melting curve analysis. Microbial quantification was based on standard curves of genomic DNA from reference strains with the LightCycler^®^ software version 4.1 (Roche Diagnostics GmbH). Data are expressed as log_10_ copies of 16S rRNA gene mL^−1^ of sample [21].

### 2.6. Prebiotic Indexes

After 24 h of fermentation of the examined substrates, Prebiotic Indexes (PIs) were calculated [24]. Calculation of Pis was based on quantification of bacteria (copies of 16S rRNA gene mL^−1^ of sample) and the following equation [38]:$$\mathrm{PI}=\left(\frac{\mathrm{Bif}}{\mathrm{Total}}\right)-\left(\frac{\mathrm{Bac}}{\mathrm{Total}}\right)+\left(\frac{\mathrm{Lac}}{\mathrm{Total}}\right)-\left(\frac{\mathrm{Clos}}{\mathrm{Total}}\right)$$where Bif is *Bifidobacterium* spp. Numbers after 24 h of fermentation/numbers at the beginning of the fermentation, Bac is *Bacteroides* spp. Numbers after 24 h of fermentation/numbers at the beginning of the fermentation, Lac is *Lactobacillus* group numbers after 24 h of fermentation/numbers at the beginning of the fermentation, Clos is *C. perfringens* group numbers after 24 h of fermentation/numbers at the beginning of the fermentation, and Total is total bacteria numbers after 24 h of fermentation/numbers at the beginning of the fermentation. 

### 2.7. Measurement of SCFAs 

Short-chain fatty acids (SCFAs) concentrations of the in vitro static batch cultures were determined by capillary gas chromatography (GC), as previously described by Mountzouris et al. [21,39]. More specifically, samples were centrifuged at 13,000× *g* for 15 min at 4 °C and 300 μL of the supernatant were stored at −80 °C until analysis. On the day of analysis, supernatants were vortexed and centrifuged at 13,000× *g* for 5 min at RT, and 85 μL of each supernatant was mixed with 10 μL 2-ethyl-butyrate (20 mM, internal standard) (2-Ethyl butyric acid 99%, Sigma-Aldrich Corp., St. Louis, MO, USA) and 5 μL hydrochloric acid (HCl, 1 M). Two microliters of each sample were injected into a gas chromatographer (Agilent 6890 GC System, Agilent Technologies, Santa Clara, CA, USA), equipped with a Stabilwax^®^-DA Capillary GC Column (size × I.D. 30 m × 0.25 mm, df 0.25 μm) (Restek Corporation, Bellefonte, PA, USA) and a flame ionization detector. Chromatography was performed with injection split ratio 35:1, injector and detector temperature set at 200 °C and 220 °C, respectively, and temperature program was run from 140 °C to 200 °C with a temperature ramp rate of 5 °C/min. Helium was the carrier gas with a column flow of 20 mL/min. Total run time was 13 min per sample. The concentrations of the SCFAs were computed based on instrument calibration with SCFA standard mix (Supelco volatile acid standard mix, Sigma–Aldrich Corp., St. Louis, MO, USA). Total short-chain fatty acids (TSCFAs) and individual SCFAs concentrations were expressed as μmol mL^−1^ of acetate, propionate, butyrate, and branched-chain SCFAs (BSCFAs; iso-butyrate, iso-valerate, iso-caproic acid), and other SCFAs (valerate, caproic acid, and heptanoic acid) were also calculated. Production rates of total VFAs, major SCFAs (acetate, propionate, butyrate) and minor SCFAs (BSCFAs, other SCFAs) after 8 h fermentation (ΔCt_8-0_, %ΔCt_8-0_) and 24 h fermentation (ΔCt_24-0_, % ΔCt_24-0_), compared to the beginning of the fermentation, were further calculated.

### 2.8. Statistical Analysis

Continuous variables are expressed as median and Q1–Q3 quartiles and categorical variables are presented as frequencies (*n*, %). For categorical variables, chi-squared (X^2^) test was applied. For continuous variables, non-parametric analysis was applied due to low number of total observations (N < 30). Bacterial levels, PIs, and SCFAs parameters between groups of ASD and neurotypical children were compared with the Mann–Whitney test. Bacterial levels, PIs, and SCFAs parameters among different treatments (NC, INU, scFOS, PO, PE) in each health group (ASD, neurotypical children) for different time points were compared with Kruskal–Wallis test with Dunn’s pairwise comparisons using the Bonferroni correction. Paired comparisons of bacterial levels and SCFAs parameters among different time points (0.8 and 24 h) for each health group (ASD, neurotypical children) and for each treatment (NC, INU, scFOS, PO, and PE) were performed by Wilcoxon signed-rank test. Statistical analysis was performed by Stata 15.1 [40]. Significance threshold was set at 5% (*p* < 0.05).

## 3. Results

### 3.1. Descriptive Characteristics of Faecal Donors

Descriptive characteristics of the faecal donors are available in Table 3. All donors were compliant with the inclusion criteria of the study concerning health status, dietary habits, or the consumption of probiotics, prebiotics, or antibiotics. No allergies or recent consumption of nonsteroidal anti-inflammatory drugs were reported among volunteers. ASD subjects and neurotypical subjects were comparable in terms of their baseline characteristics, including sociodemographic factors, anthropometric indices, and dietary intake. Regarding stool characteristics, the ASD children exhibited a tendency for more frequent evacuation times per day, while the same group displayed significantly more acidic stool’s pH levels compared to neurotypical children. Additionally, higher faecal moisture and score ratings according to Bristol Stool Scale (BSS) were detected in ASD subjects compared to neurotypical subjects, without reaching statistical significance. 

### 3.2. Faecal Microbiota Analysis 

#### 3.2.1. Bacterial Enumeration in Neurotypical Children’s Faeces after 24 h of Fermentation

The outcome of the gut microbiota analysis after 24 h of fermentation, for each examined substrate in five neurotypical children is presented in Table 4. In vitro fermentation with NC resulted in significantly decreased levels of the butyrate producer *F. prausnitzii* (*p* = 0.043) compared to the beginning of the fermentation (Table 4 and Appendix A). Both prebiotics INU and scFOS induced a significant increase in *Bifidobacterium* spp. levels compared to NC (*p* = 0.004 and *p* = 0.040, respectively) and compared to t = 0 (*p* for all = 0.043). Moreover, a significant percentage median change in *Bifidobacterium* spp. levels was also observed, for both positive controls, compared to NC (*p* = 0.006 and *p* = 0.020, respectively, Appendix A). In addition, 24 h of fermentation of the known prebiotics significantly increased the levels of total bacteria and *Bacteroides* spp. compared to the beginning of the fermentation (*p* for all = 0.043) (Table 4 and Appendix A). Fermentation of *P. eryngii* and *P. ostreatus* mushrooms resulted in significantly augmented levels of total bacteria compared to NC (*p* = 0.017 and *p* = 0.026, respectively) and compared to t = 0 (*p* for all = 0.043). *Bifidobacterium* spp. and *Bacteroides* spp. levels were significantly increased for both mushrooms after 24 h of fermentation, compared to t = 0 (*p* for all = 0.043). Moreover, PO induced a significant increase in *F. prausnitzii* levels (9.08 (8.36–9.20)), while 24 h fermentation of PE resulted in elevated levels of *A. muciniphila* (4.67 (4.45–8.32)) compared to the beginning of the fermentation (*p* for all = 0.043). Finally, no significant differences were observed regarding the levels of *Lactobacillus* group, *C. perfrigens* group, and *Prevotella* spp., for all the examined substrates. 

#### 3.2.2. Bacterial Enumeration in ASD Children’s Faeces after 24 h of Fermentation

Table 5 demonstrates gut microbiota analysis, after 24 h of fermentation, for each examined substrate in five autistic children. In detail, NC was characterised by increased levels of *Bifidobacterium* spp. (*p* = 0.043), after 24 h of fermentation compared to the beginning of the fermentation (Appendix A). In addition, total bacteria, *Bifidobacterium* spp., *Bacteroides* spp., and *Prevotella* spp. were significantly elevated for both positive controls, INU and scFOS, after 24 h of fermentation compared to the beginning of fermentation (*p* for all = 0.043) (Table 5 and Appendix A). Based on our experimental data, 24 h of fermentation of PO was characterised by higher *Bifidobacterium* spp. and *Prevotella* spp. levels, compared to t = 0 (*p* for all = 0.043), while a similar pattern was detected in the case of PE mushroom (*p* for all = 0.043). Moreover, PE mushroom significantly elevated the levels of total bacteria, *Bacteroides* spp., and *F. prausnitzii*, after 24 h of fermentation compared to NC (*p* = 0.002, *p* = 0.011 and *p* = 0.030, respectively). Finally, no significant differences were observed regarding the levels of *Lactobacillus* group, *C. perfringens* group, and *A. muciniphila* for all the examined substrates.

#### 3.2.3. Alterations in Faecal Bacterial Populations between ASD and Neurotypical Children after 24 h of Fermentation 

After 24 h of fermentation with NC, total bacteria significantly increased in ASD children compared to neurotypical children (10.14 (10.05–10.18) vs. 9.83 (9.77–9.99), respectively, *p* = 0.016) (Table 4 and Table 5), while a similar trend for the percentage changes from t = 0 of *Prevotella* spp. levels (*p* = 0.050) was also observed (Appendix A). Both positive controls, INU and scFOS, induced increased levels of *F. prausnitzii* in autistic children (9.03 (8.80–9.20) and 8.92 (8.82–9.16), respectively), compared to neurotypical subjects (8.67 (8.11–8.88) and 8.62, respectively), after 24 h of fermentation (*p* for all = 0.028) (Table 4 and Table 5). In addition, a similar trend was observed for the percentage changes of *F. prausnitzii*, especially for INU (*p* = 0.095) (Appendix A). No significant differences were detected between ASD and neurotypical children regarding the 24 h fermentation of PO mushroom. In the case of PE mushrooms, autistic children had a trend for reduced levels of *Prevotella* spp. after 24 h of fermentation compared to healthy subjects (7.32 (7.15–8.63) vs. 9.30 (8.25–9.98), respectively, *p* = 0.086) (Table 4 and Table 5). Finally, the percentage changes from the beginning of the fermentation of *Prevotella* spp. during fermentation of PE mushroom were higher in ASD children compared to neurotypical subjects (*p* = 0.050) (Appendix A).

### 3.3. Prebiotic Indexes (PIs)

Figure 1 demonstrates prebiotic indexes after 24 h of fermentation of the tested substrates. The prebiotic index offers the advantage of normalising the bacterial population alterations in relation to the initial microbial levels, accounting for the physiological variability that characterises the experimental process of in vitro fermentation [38]. According to our results, in both groups, no significant differences among treatments were detected. A trend for higher PI for the scFOS treatment compared to PO mushroom was detected in neurotypical children (*p* = 0.099). The calculation of prebiotic indexes per subject for each one of the treatments revealed the variability among subjects (Appendix A). More specifically, the NC of each donor regarding the ASD group induced negative PIs in half of the volunteers, while the other donors induced low values of the prebiotic indexes (PIs of the ASD group were calculated in four of the five participating volunteers, as *C. perfringens* was not detected in the 3rd donor). On the contrary, the neurotypical group induced positive PIs in most volunteers. Moreover, the 24 h fermentation of prebiotic inulin and scFOS induced positive PIs only in all five neurotypical children. 

### 3.4. SCFA Production 

#### 3.4.1. SCFA Production after 8 and 24 h of Fermentation for the Neurotypical Children Group

Table 6 demonstrates SCFAs production per treatment in the group of non-autistic children after 8 and 24 h of fermentation. In healthy subjects’ faecal samples at 8 h of fermentation, NC exhibited significantly increased levels of TSCFAs, acetate, propionate, BSCFAs, and other SCFAs, compared to t = 0 (*p* for all = 0.043) (Appendix A). After 24 h of fermentation, for the same treatment, significant concentration increments of the major SCFAs (acetate: 13.06 (11.46–13.61), propionate: 2.77 (2.47–5.05) and butyrate: 2.70 (1.95–3.59) μmol mL^−1^), BSCFAs (2.39 (2.11–2.50) μmol mL^−1^) and other SCFAs (1.17 (0.74–1.25) μmol mL^−1^) were noticed, compared to the beginning and to 8 h of fermentation (*p* for all = 0.043). 

Positive control INU demonstrated significantly increased levels of acetate, propionate, TSCFAs and other SCFAs after 8 h of fermentation, compared to t = 0 (*p* for all 0.043) (Appendix A). After 24 h of fermentation, INU treatment was characterised by a significant increase in major SCFAs (acetate: 45.37 ± 18.43 μmol mL^−1^, propionate: 11.55 ± 7.51 μmol mL^−1^, butyrate: 6.91 ± 5.07 μmol mL^−1^), TSCFAs (64.44 ± 16.25 μmol mL^−1^), and other SCFAs (0.34 ± 0.12 μmol mL^−1^) concentrations, compared to t = 0 and to 8 h of fermentation (*p* for all = 0.043). Butyrate concentration was significantly increased (4.97 (3.95–10.85) μmol mL^−1^) after 24 h of fermentation, compared to t = 0 and to 8 h of fermentation (*p* for all = 0.043). Additionally, a reduced production of BSCFAs was observed compared to NC (*p* = 0.023) (Appendix A). 

In the case of the scFOS, significant increments were recorded for major SCFAs and TSCFAs after 8 h and 24 h of fermentation, compared to t = 0 (*p* for all = 0.043) (Table 6 and Appendix A). Furthermore, acetate concentrations (8 h: 25.57 (15.70–35.04) μmol mL^−1^, *p* = 0.030; 24 h: 67.79 (52.66–68.99) μmol mL^−1^, *p* = 0.001) and production rates (Appendix A) were significantly elevated compared to NC and to 8 h of fermentation (*p* for all = 0.043). Moreover, propionate (5.18 (3.85–14.23) mol mL^−1^, *p* = 0.043) and TSCFAs (73.51 (71.48–79.49) mol mL^−1^, *p* = 0.043) levels increased significantly between 8 and 24 h of fermentation. Branched and other SCFAs production was significantly decreased after 24 h of fermentation, compared to NC (*p* = 0.008 and *p* = 0.005, respectively), and to 8 h of fermentation in the case of other SCFAs (*p* = 0.043). 

PO fermentation for 8 h was characterised by an increase in major SCFA concentrations, including BSCFAs, other SCFAs, and TSCFAs, compared to t = 0 (*p* for all = 0.043) (Table 6 and Appendix A). For the same time point, fermentation of PO induced a significant higher production of butyrate (% ΔCt_8-0_: 1005.29 (772.94–2188.18) μmol mL^−1^, *p* = 0.015) compared to NC, with a similar trend in the case of TSCFAs (% ΔCt_8-0_: 1784.06 (1180.82–1940.49), ΔCt_8-0_: 31.34 (20.55–44.28), 33.79 (22.03–46.65) μmol mL^−1^) and propionate (% ΔCt_8-0_: 1123.70 (847.83–2365.36), *p* = 0.060) (Appendix A). After 24 h of fermentation, PO treatment was characterised by an increase in major SCFAs (acetate 38.90 (33.21–39.80) μmol mL^−1^, propionate: 11.19 (8.76–19.98) μmol mL^−1^, butyrate: 14.15 (10.39–17.52) μmol mL^−1^) and TSCFAs (71.03 (57.36–74.97) μmol mL^−1^) concentrations compared to t = 0 (*p* for all = 0.043), while the increase in propionate and TSCFAs was also significant compared to 8 h of fermentation (*p* for all = 0.043). BSCFAs and other SCFAS levels were significantly lower after 24 h of fermentation, compared to t = 0 and to 8 h of fermentation (*p* for all = 0.043). 

In healthy subjects, 8 h of fermentation with PE mushroom was characterised by an increase in major SCFAs and TVFAs compared to t = 0 (*p* for all = 0,043), while levels of acetate and TSCFAs were significantly increased compared to NC (*p* = 0.046 and *p* = 0.020, respectively) (Table 6 and Appendix A). After 24 h of fermentation, PE mushroom induced a significant increase in propionate (16.53 (9.92–27.63) μmol mL^−1^), butyrate (16.45 (14.34–21.18) μmol mL^−1^), and TSCFAs (81.38 (76.27–84.40) μmol mL^−1^) levels compared to t=0 (*p* for all = 0,043), NC (*p* = 0.015, *p* = 0.004 and *p* = 0.002, respectively) and to 8 h of fermentation (*p* for all = 0,043). Furthermore, 24 h of fermentation with PE resulted in significantly higher production of TSCFAs (ΔC_t24-0_: 79.62 (74.63–81.83) μmol mL^−1^, % ΔC_t24-0_: 4208.17 (2859.96–6387.16) μmol mL^−1^), propionate (ΔC_t24-0_: 16.28 (9.65–27.00) μmol mL^−1^, % ΔC_t24-0_: 6442.25 (2145.62–7913.05) μmol mL^−1^), and butyrate (ΔC_t24-0_: 16.07 (14.18–20.76) μmol mL^−1^, % ΔC_t24-0_: 5569.62 (4012.14–10355.36) μmol mL^−1^) compared to NC, with butyrate changing significantly more than scFOS (*p* = 0.023) (Appendix A). 

Overall, both prebiotics (INU and scFOS) induced a significant increment of major SCFAs after 24 h of fermentation, whereas scFOS resulted in significantly increased levels and production rates of acetate compared to NC. Finally, *P. eryngii* mushrooms induced a significant increase in levels and production rates of TSCFAs, propionic acid, and butyrate after 24 h of fermentation compared to NC.

#### 3.4.2. SCFA Production after 8 and 24 h of Fermentation for the Autistic Children Group

Table 7 demonstrates SCFAs production per treatment in the autistic children group after 8 and 24 h of fermentation. In the faecal samples of the autistic group, NC treatment was characterised by a significant increase in acetate, propionate, and TSCFAs after 8 and 24 h of fermentation compared to t = 0 (Appendix A) baseline, also posing a significant increment between 8 h and 24 h fermentation (*p* for all = 0.043). Butyrate levels (3.34 (2.81–4.91) μmol mL^−1^) were also significantly increased after 24 h of fermentation, compared to t = 0 and to 8 h of fermentation (*p* for all = 0.043). Branched and other SCFA production was significantly elevated after 24 h of fermentation, compared to t = 0 and to 8 h of fermentation (*p* for all = 0.043). 

In the case of the prebiotic INU, significant increments were recorded for major SCFAs, TSCFA, and BSCFAs after 8 h and 24 h of fermentation compared to t = 0 (*p* for all = 0.043) (Table 7 and Appendix A). Additionally, concerning 24 h of fermentation, the aforementioned SCFAs demonstrated a significant increase compared to 8 h of fermentation (*p* for all = 0.043). Other SCFAs production was found to be significantly elevated after 24 h of fermentation, compared to t = 0 and to 8 h of fermentation (*p* for all = 0.043). 

The 8 h fermentation process resulted in significant concentration increments of the major SCFAs and TSCFAs in scFOS treatment, compared to t = 0 (*p* for all = 0.043) (Table 7 and Appendix A). In addition, a significantly greater production of TVFAs (% ΔC_t8-0_: 1240.45 (835.68–1718.86 μmol mL^−1^), *p* = 0.046) and acetate (%ΔC_t8-0_: 1751.66 (1032.01–2558.34) μmol mL^−1^, *p* = 0.026) was observed, compared to NC (Appendix A). Furthermore, 8 h of fermentation of scFOS resulted in a significantly lower concentration of BSCFAs, compared to t = 0 (*p* for all = 0.043) and NC (*p* = 0.020). After 24 h of fermentation, acetate (52.91 (34.03–70.19) μmol mL^−1^), propionate (3.87 (2.39–6.24) μmol mL^−1^), and TSCFAs (63.78 (46.38–79.41) μmol mL^−1^) exhibited a significant increase compared to t = 0 as well as to 8 h of fermentation (*p* for all = 0.043), whereas acetate levels were also significantly elevated compared to NC (*p* = 0.009). Moreover, butyrate levels (6.10 (2.54–9.75) μmol mL^−1^) were significantly increased after 24 h of fermentation, compared to t = 0 (*p* = 0.043). The production of the other SCFAs and BSCFAs was significantly decreased compared to NC in the autistic group (*p* = 0.011 and *p* = 0.001, respectively), after 24 h of fermentation with prebiotic scFOS. 

Fermentation of PO mushrooms resulted in significantly higher levels of major SCFAs, TSCFAs, and BSCFAs after 8 h and 24 h of fermentation, compared to t = 0 (*p* for all = 0.043) (Table 7 and Appendix A). Furthermore, when compared to NC, 8 h of fermentation resulted in significantly greater production of TSCFAs (%ΔC_t8-0_: 1239.59 (971.88–1654.21) μmol mL^−1^, *p* = 0.046), propionate (%ΔC_t8-0_: 1719.78 (929.90–2070.08) μmol mL^−1^, *p* = 0.020), ΔC_t8-0_: 0.60 (3.30–7.03 μmol mL^−1^, *p* = 0.060), and butyrate (ΔC_t8-0_: 7.11 (3.01–9.62) μmol mL^−1^, *p* = 0.030 and %ΔC_t8-0_: 1227.34 (996.60–2214.84), *p* = 0.09) (Appendix A). Furthermore, after 24 h of fermentation, propionate (*p* = 0.040) and butyrate (*p* = 0.035) concentrations as well as production rates (acetate: %ΔC_t24-0_ and ΔC_t24-0_, *p* = 0.030 and *p* = 0.09, respectively; butyrate: %ΔC_t24-0_ and ΔC_t24-0_, *p* = 0.015 and *p* = 0.06, respectively) were significantly increased compared to NC (Appendix A). Moreover, between 8 h and 24 h of fermentation with PO mushroom, elevated levels of major SCFAs, TSCFAs, and BSCFAs were also observed (*p* for all = 0.043). 

PE treatment was characterised by an increase in major SCFAs, TSCFAs, and BSCFAs concentrations after 8 h and 24 h of fermentation compared to t = 0 (*p* for all = 0.043) (Table 7 and Appendix A). More specifically, at 8 h of fermentation, PE led to significantly increased levels of acetate (*p* = 0.013), propionate (*p* = 0.035), and TSCFAs (*p* = 0.004), compared to NC, with a similar trend for butyrate (*p* = 0.068). Additionally, for the same time point, PE fermentation induced a significantly elevated production (ΔC_t24-0_ and %ΔC _t24-0_) of TSCFAs (*p* = 0.002, *p* = 0.003), acetate (*p* for all = 0.009), propionate (*p* = 0.023, *p* = 0.005), and butyrate (*p* = 0.035, *p* = 0.011), compared to NC (Appendix A). After 24 h of fermentation, the concentrations of propionate (12.63 (9.79–17.90) μmol mL^−1^, *p* = 0.008), butyrate (19.97 (7.35–26.23) μmol mL^−1^, *p* = 0.040), and TSCFAs (78.61 (60.99–84.45) μmol mL^−1^, *p* = 0.002) in PE mushroom were significantly higher than in NC, with a similar trend for acetate (*p* = 0.052). In addition, compared to NC, there was a significantly higher production (Δ and %Δ) of TSCFAs (*p* = 0.002, *p* = 0.004), acetate (*p* = 0.040, *p* = 0.030), propionate (*p* = 0.006, *p* = 0.002), and butyrate (*p* = 0.017, *p* = 0.003) (Appendix A). Moreover, in PE treatment, there was a trend for lower production of BSCFAs compared to NC after 8 h of fermentation. (*p* = 0.009). 

Overall, both prebiotic controls induced a significant increase in major SCFAs compared to the beginning of the fermentation, whereas scFOS caused a significant increase in acetate, propionate, and TSCFAs compared to NC after 24 h of fermentation. Finally, both mushrooms exhibited a significant increment in levels and production rates of propionate and butyrate, compared to NC after 24 h of fermentation.

#### 3.4.3. Differences in Metabolic Products between ASD and Neurotypical Children’ Faeces after 24 h of Fermentation 

Mann–Whitney test analysis was applied between the ASD and neurotypical children for each treatment in order to determine any metabolic differences between the two groups. In NC treatment, the neurotypical group induced significantly higher percentage production of ΔTVFAs (%ΔC_t24-0_: 534.15 (464.00–653.35) μmol/mL^−1^ vs. 348.20 (250.87–418.93) μmol/mL^−1^) compared to the ASD group after 24 h of fermentation, though the ASD group exhibited a trend for higher TSCFAs levels at the end of fermentation (*p* = 0.076) (Appendix A). Moreover, in the same treatment, the neurotypical group exhibited a trend for higher production of acetate, whereas the ASD group demonstrated a trend for higher iso-caproic concentration (*p* for all = 0.076). Furthermore, 24 h of fermentation with prebiotic inulin resulted in a significantly higher production of propionate (%ΔC_t24-0_: 2184.12 (1680.62–3658.01) μmol/mL^−1^ vs. 1082.30 (647.15–1376.36) μmol/mL^−1^, *p* = 0.028) and a trend for higher production of ΔTSCFAs (*p* = 0.056) and acetate (*p* = 0.095) in neurotypical group compared to ASD children (Appendix A). On the other hand, ASD children exhibited significantly higher BSCFA production rates (*p* = 0.0016 and *p* = 0.009) after inulin fermentation compared to the healthy group (isobutyrate concentration: *p* = 0.016 and iso-caproate concentration *p* = 0.009). In scFOS treatment, no significant differences were detected. Nevertheless, like inulin, the neurotypical group had a trend for higher percentage production rate of ΔTSCFAs (*p* = 0.076) and acetate (*p* = 0.056) compared to ASD after 24 h of fermentation (Appendix A). Regarding the PO treatment, the neurotypical group demonstrated significantly higher percentage production rate of acetate (*p* = 0.009) and a trend for a higher % ΔTSCFAs, Δacetate, and other SCFAs characteristics compared to ASD after 24 h of fermentation (*p* = 0.095, *p* = 0.056 and *p* = 0.095, respectively) (Appendix A). In PE treatment, no significant differences were detected. Nevertheless, like other treatments, the neurotypical group had a trend for a higher % Δacetate (*p* = 0.076) compared to ASD after 24 h of fermentation (Appendix A). Thus, the most consistent trait was the higher % Δacetate in neurotypical children in all treatments. Finally, fermentation of inulin induced higher BSCFAs characteristics in ASD compared to the healthy group.

## 4. Discussion

The pathogenesis of ASD is complex, and in addition to the genetic background, other environmental factors such as the gut microbiota may play a key role in the symptomology of ASD. There is still more research required to fully understand how gut microbiota increases an individual’s risk of developing autism as well as the evidence connecting the ASD symptoms to gut dysbiosis [41]. The frequent occurrence of GI symptoms in ASD children, however, raises the possibility that the gut microbiota is involved in GI pathology, providing a possible target for diagnostic and therapeutic approaches. Recent studies have focused on the impact of pre/probiotics on the gut–brain axis [42,43]. However, evidence for the possible effect of prebiotics on ASD is lacking. The purpose of this study was to examine the effects of edible mushrooms (potential prebiotics) as well as known prebiotic compounds (inulin and scFOS) on the gut microbiota composition and metabolite production in vitro, using faecal samples from autistic and non-autistic children.

Microbiota-based strategies have highlighted their potential to regulate CNS-driven behaviors [44,45]. In recent years, several studies have shown that administration of probiotics and prebiotics can be a potential therapeutic strategy for treating neurological disorders [46,47,48,49]. A new strategy for treating ASD may involve dietary changes, probiotic or prebiotic treatment, microbiota transfer therapy, or targeted antibiotic therapy. However, the evidence regarding prebiotics is limited. Three clinical and two in vitro studies have been conducted to date, with different types of prebiotic compounds, such as partially hydrolyzed guar gum [50], galactooligosaccharides [16,27,46], and fructo-oligosaccharides combined with probiotics [51], related to autism spectrum disorders. To our knowledge, this is the first study to examine the effects of edible mushrooms on gut-faecal microbiota characteristics of ASD children. In the present study, we enumerated specific members of gut microbiota that, according to other in vitro studies, may be involved in the pathogenesis of ASD [16,27]. In addition, several studies have highlighted that in vitro fermentation with prebiotic compounds can modify the composition of the selected gut microbiota members [21,52]. In our study, 24 h fermentation of edible mushrooms *P. eryngii* and *P. ostreatus* significantly increased the levels of *Bifidobacterium* in both groups. Although alterations in gut microbiota in ASD children are not always consistent among studies, patients frequently have reduced levels of *Bifidobacterium* compared to non-autistic children [17,53,54]. Ahmed et al. [55] revealed that the only significant alteration between the GM of ASD children and that of their healthy siblings was the higher abundance of *Bifidobacterium* in the siblings’ group, supporting its protective role. Several studies have reported various species of *Bifidobacterium* as GABA (gamma aminobutyric acid) producers [56,57]. GABA is the main inhibitory neurotransmitter in the brain, the concentrations of which have been found to be diminished in ASD children [58,59]. Recently, we demonstrated that in vitro fermentation of the aforementioned mushrooms promoted the growth of *Bifidobacterium* spp. using faecal inocula from elderly donors, highlighting their beneficial effects on gut microbiota dynamics during the aging process [21]. This bifidogenic effect was demonstrated by another in vitro study using faecal inocula from healthy donors, where the edible mushrooms, including *P. eryngii* and *P. ostreatus*, promoted the growth of *Bifidobacterium* [60]. Regarding the autism spectrum disorders, this effect was analogous to the known prebiotic GOS, as it was demonstrated in two in vitro studies [27,46].

Bifidobacteria are considered important acetate producers, and they participate in cross-feeding interactions with other gut bacterial groups. *P. eryngii* and *P. ostreatus* mushrooms may have bifidogenic properties due to their high β-glucan content. According to previous results, the content of β-glucans was 38.7 ± 5.4 PE and 30.6 ± 1.9 (%, *w*/*w*, d.w.) for PE and PO mushrooms, respectively [21]. Various studies have demonstrated that β-glucan improves growth, viability, and colonization of probiotic microorganisms such as *Bifidobacterium* [61,62]. Besides the bifidogenic effect of both mushrooms, PO resulted in elevated levels of *F. prausnitzii* in neurotypical children’s faecal samples, while PE mushrooms led to an increase in *A. muciniphila*, compared to the beginning of the fermentation. In the ASD group, only PE mushroom resulted in significantly elevated levels of total bacteria, *Bacteroides*, and *F. prausnitzii* compared to NC. Those differences could be attributed to the varying mushroom properties/content of the two species examined. 

Regarding the known prebiotic compounds, their 24 h fermentation led to alterations involving different microbial populations in relation to the effect of mushrooms. More specifically, inulin and scFOS significantly increased the levels of total bacteria and *Bacteroides* in both groups, while inulin resulted in significantly elevated levels of *Bifidobacterium* in neurotypical children and of *Prevotella* spp. in the ASD group. Notably, the comparison between the two groups resulted in elevated levels of *F. prausnitzii* in the ASD group for both prebiotics. *F. prausnitzii* is one of the most dominant commensal and butyrate-producing species in the human gut, which has been selectively modulated in order to restore the gut microbiota composition and enhance intestinal health [63]. Various clinical trials have reported that diets enriched with fructo-oligosaccharides (FOS) and inulin increase *F. prausnitzii* levels and, in turn, provide multiple carbon and energy sources to the colonic epithelium and improve gut health [64]. In addition, the levels of the colonic butyrate-producing bacteria have been found to be decreased in ASD children [16]. 

Emerging evidence indicates that the composition and abundance of SCFAs differ in children with ASD, and contribute to the etiology of ASD. Although the underlying mechanisms through which SCFAs might influence brain physiology and behavior have not been fully elucidated, they include alterations in neurotransmitter production, mitochondrial function, immune activation, lipid metabolism, and gene expression [15,65,66]. 

In our study, the fermentation process resulted in significant concentration increments of the major SCFAs and TVFAs in prebiotic treatments for both groups. Inulin and scFOS are well-known prebiotics that are used selectively by gut microbiota and are generally associated with the stimulation of beneficial bacteria, which results in the production of SCFAs. The increase in SCFA production by prebiotic treatments noted herein can be considered positive. This is likely due not only to a compromised gut microbiota in ASD children (such as reduced levels of *Bifidobacterium*) [53], but also to decreased metabolic activity of intestinal bacteria [27]. Notably, 24 h of fermentation with scFOS resulted in increased levels of acetate compared to NC. Acetate is the most abundant SCFA in the colon and bloodstream, and is produced by a vast number of anaerobic bacteria, including species of the genera *Prevotella*, *Bifidobacterium*, and *Ruminococcus* [67]. Although there are differences in bacterial abundances in ASD, these genera have been reported to be reduced in ASD individuals [17,68]. Grimaldi et al. [16] demonstrated that administering B-GOS in an in vitro model simulating the gut of autistic children resulted in increased concentrations of acetate and butyrate. More recently, Wang et al. [51] observed a significant increase in SCFAs in children with ASD after an intervention combining probiotics and fructo-oligosaccharides.

Similar to what was observed in the case of the prebiotic compounds, the 24 h fermentation of all mushroom-based substrates induced the production of major SCFAs and TSCFAs for both groups, except for acetate after fermentation with PE mushroom in neurotypical children. Notably, both mushrooms induced elevated levels of butyrate in ASD group compared to NC, after 24 h of fermentation. Butyrate can be absorbed by the colonic mucosa and serve as an energy substrate for intestinal epithelial cells, while it also exhibits anti-cancer, pro-apoptotic, and anti-inflammatory properties [21]. Butyrate modulates the biosynthesis of catecholamines and neurotransmitters in the central and autonomic nervous system, as well as promoting memory formation and neuronal plasticity through epigenetic modulation [69]. Butyrate has also been shown to alleviate symptoms in some neurodegenerative models [70]. Even if not conclusive, the above data highlight that decreased butyrate levels may affect the pathophysiology of ASD. Regarding the healthy group, in the case of both mushrooms, we did not notice an outstanding increase in the tested butyrate producers. On the contrary, significantly increased levels of *F. prausnitzii* were recorded after 24 h of fermentation in the ASD group, compared to neurotypical children group. Future studies will include DNA (e.g., 16S rRNA) sequencing of the gut microbiota examined in order to obtain more information about other, non-tested, butyrate producers.

Propionic acid is one of the most abundant SCFAs produced by *Clostridium*, *Bacteroides*, and *Desulfovibrio* species, and has been of key interest because it has several links to autism spectrum disorder. Propionate may be beneficial at appropriate levels and exert many useful functions, including neurological, metabolic, and physiological function, tumour suppression, and anti-inflammation [71,72]. Propionate has been reported to be upregulated in people with ASD and associated with increased severity of ASD; however, it remains to be determined whether enhanced levels in the intestine are high enough to generate a notable effect in the brain [13]. Excessive propionate levels, as reflected in rat model studies, have revealed behavioral, metabolic, and neuropathological changes as well as neuroinflammation and increased oxidative stress, which are all consistent with common symptoms related to autism spectrum disorders [73]. In our study, we did not find significant differences of faecal propionate in autistic subjects (compared to neurotypical children) prior to the fermentation process, whereas 24 h fermentation of both mushrooms resulted in significantly elevated levels of propionate in the ASD group compared to the NC. 

Overall, the in vitro results of the present study suggest that addition of the edible mushrooms *P. eryngii* and *P. ostreatus* to the diet—and consequently to the gut microbiota—of children with ASD may regulate the gut bacterial population and affect metabolic activity in a way that could be advantageous to the host’s health. In order to overcome the limitations mainly set by the extent of microbiome analysis, we intend to expand our study by implementing an intervention study and gut microbiota sequencing (e.g., 16S rRNA), in order to obtain more information about other, non-tested, gut microbiota members.

## 5. Conclusions

Recent data demonstrated that GM manipulation could be a promising strategy in the prevention and/or treatment of neurological diseases, including autism spectrum disorders. Edible mushrooms are inexpensive, safe food choices, and their potential positive effects on human health have been recently highlighted. However, data from the in vitro fermentation of edible mushrooms using faecal samples derived from ASD individuals are missing. The present work aimed to investigate the effects of *P. eryngii* and *P. ostreatus* mushrooms as potential novel prebiotics with possible beneficial effects on ASD children. An in vitro fermentation study was performed using fresh faecal inocula from five autistic child donors and five age-matched typically developing controls. We used lyophilized powder from the entire fruitbodies of mushrooms in order to explore the possible synergistic effects of their constituents, instead of focusing only on a single category of potentially bioactive substances (e.g., β-glucans). Our data suggested that edible mushrooms may manipulate in vitro the gut bacterial population and metabolic activity, especially in the ASD group. Fermentation of both mushrooms significantly increased the levels of *Bifidobacterium*, while *P. eryngii* mushrooms resulted in significantly elevated levels of *F. prausnitzii* in the ASD group compared to NC, accompanied by a substantial increase in butyrate production. Further in vivo and human studies are necessary to elucidate the effects of edible mushrooms on autism spectrum disorders, in order to generate effective interventions for individuals and offer novel targeted therapeutics.

## Figures and Tables

**Figure 1 microorganisms-11-00414-f001:**
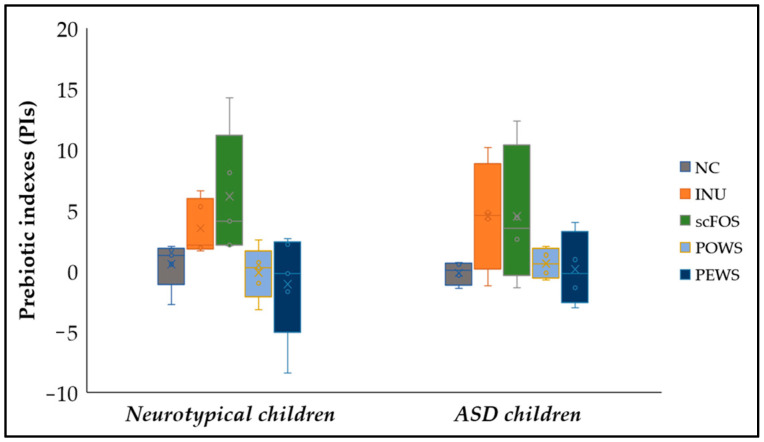
Prebiotic indexes for each substrate in autistic and neurotypical children. Data are presented as Boxplots, with the horizontal line representing the median and the whiskers representing the minimum and maximum values.

**Table 1 microorganisms-11-00414-t001:** Substrates used in the experiments.

Description	Abbreviation
*Pleurotus eryngii* LGAM 216	PE
*Pleurotus ostreatus* IK 1123	PO
Inulin	INU
Short-chain fructooligosaccharides	scFOS
Negative control (basal medium with no carbohydrate source)	NC

**Table 2 microorganisms-11-00414-t002:** Species-, genus-, and group-specific primers targeting the 16S rRNA and qPCR characteristics of gut microbiota analysis.

Target	Primer	Primer Sequence (5′-3′)	Reference Strains	References
Total Bacteria	Forward	TCCTACGGGAGGCAGCAGT	*Bacteroides fragilis* MM44 (ATCC 25285)	[28]
	Reverse	GGACTACCAGGGTATCTAATCC TGTT		
*Lactobacillus* group	Forward	AGCAGTAGGGAATCTTCCA	*L. gasseri* DSM 20243	[29]
	Reverse	CACCGCTACACATGGAG		
*Bifidobacterium* spp.	Forward	TCGCGTCYGGTGTGAAAG	*B. bifidum* DSM 20456	[29]
	Reverse	CCACATCCAGCRTCCAC		
*F. prausnitzii*	FPR-2F	GGAGGAAGAAGGTCTTCGG	*F. prausnitzii* DSM 17677	[30,31,32]
	Fprau645R	AATTCCGCCTACCTCTGCACT		
*A. muciniphila*	AM1	CAGCACGTGAAGGTGGGGAC	*A. muciniphila* DSM 22959	[33,34]
	AM2	CCTTGCGGTTGGCTTCAGAT		
*C. perfringens* group	CPF	ATGCAAGTCGAGCGATG	*C. perfringens* ATCC 13124	[35]
	CPR	TATGCGGTATTAATCTCCCTTT		
*Prevotella* spp.	g-Prevo-F	CACRGTAAACGATGGATGC	*Prevotella copri* DSM 18205	[36]
	g-Prevo-R	GGTCGGGTTGCAGACC		
*Bacteroides* spp.	Bac303F	GAAGGTCCCCCACATTG	*B*. *fragilis* MM44 (ATCC 25285)	[37]
	Bfr-Fmrev	CGCKACTTGGCTGGTTCAG		

**Table 3 microorganisms-11-00414-t003:** Descriptive characteristics of fecal donors (n = 10).

	ASD Subjects (n = 5)	NT Subjects (n = 5)	*p*-Value
Sociodemographic parameters			
Sex (no. of males/females), n (%)	5/0 (100%, 0%)	5/0 (100%, 0%)	1.000
Age (years)	5.5 (5.5, 6)	7 (5, 7.5)	0.461
Stool characteristics			
Evacuation frequency (times d^−1^)	2 (1.5, 2)	1 (1, 1.5)	0.072
Stool moisture (%)	75.97 (68, 31, 85, 18)	63.19 (59.51, 78.40)	0.175
Stool pH	5.74 (5.66, 6.25)	7.04 (6.32, 7.41)	0.016 *
Bristol Stool Scale	4 (3, 4.5)	2 (1.5, 4.5)	0.167
Anthropometric measurements			
Body weight (kg)	25 (21, 26)	26 (14.6, 28)	0.917
Height (cm)	110 (110, 125)	124 (112, 129)	0.462
BMI (kg m^−2^)	17.3 (16, 17.4)	15.6 (15.3, 16.8)	0.465
Nutritional analysis			
Energy intake (kcal d^−1^)	1742.5 (1645, 1909.2)	1435.5 (1092.6, 1694.3)	0.463
Carbohydrate (% of energy)	40.2 (39.4, 40.2)	45.4 (44.5, 48.5)	0.075
Carbohydrate (g d^−1^)	170.5 (163.9, 191.7)	146 (134.3, 192.4)	0.600
Protein (% of energy)	19.6 (17, 22.9)	17.6 (17.1, 23.9)	0.600
Protein (g d^−1^)	80.7 (62.9, 109)	66.2 (57.2, 74.4)	0.600
Fat (% of energy)	36.9 (36.9, 41)	30.7 (27.6, 37.1)	0.463
Fat (g d^−1^)	78.3 (75.1, 78.3)	69.8 (34, 70.5)	0.075
Fibers (g d^−1^)	10.7 (9.7, 11.1)	9.6 (8.4, 13.6)	0.917

Values are expressed as mean and SD for parametric or median and Q1–Q3 quartiles for nonparametric data; BMI: Body Mass Index, ASD: Autism spectrum disorder; NT: Neurotypical; * significantly different between groups (*p* < 0.05) (Mann–Whitney test).

**Table 4 microorganisms-11-00414-t004:** Faecal microbial quantification after 24 h of fermentation in neurotypical children.

	NC	INU	scFOS	PO	PE	Overall *p*
Total bacteria	9.83 (9.77–9.99)	10.34 (10.26–10.37) ^a^	10.33 (10.27–10.41) ^a^	10.40 (10.27–10.47) *^,a^	10.41 (10.30–10.60) *^,a^	0.013
*Bifidobacterium* spp.	8.96 (8.60–9.47)	10.17 (9.88–10.30) *^,a^	10.23 (10.15–10.33) *^,a^	9.90 (9.34–10.00) ^a^	9.72 (9.23–10.18) ^a^	0.004
*Lactobacillus* group	5.09 (4.76–6.31)	5.43 (4.75–6.40)	5.31 (4.61–6.35)	5.35 (4.80–6.29)	5.21 (4.88–6.52)	0.998
*C*. *perfringens* group	6.99 (6.54–7.21)	7.27 (6.74–7.93)	7.18 (6.70–7.79)	7.25 (6.69–7.56)	7.42 (6.79–8.03)	0.563
*Bacteroides* spp.	9.16 (9.04–9.44)	9.85 (9.73–10.10) ^a^	9.67 (9.59–10.08) ^a^	10.00 (9.60–10.28) ^a^	9.82 (9.51–10.21) ^a^	0.033
*F*. *prausnitzii*	8.24 (6.97–8.49) ^a^	8.67 (8.11–8.88)	8.62 (8.07–8.86)	9.08 (8.36–9.20) ^a^	8.80 (8.52–9.25)	0.050
*Prevotella* spp.	7.69 (7.44–8.52)	9.61 (7.87–10.05)	9.39 (7.83–9.92)	9.35 (7.12–10.09)	9.30 (8.25–9.98)	0.431
*A*. *muciniphila*	4.80 (4.65–8.23)	4.81 (4.25–7.99)	4.76 (4.18–8.20)	4.74 (4.55–8.26)	4.67 (4.45–8.32) ^a^	0.981

Values are expressed as median and Q1–Q3 quartiles for non-parametric data; ^a^: significantly different compared to t = 0 h for neurotypical children (Wilcoxon signed-rank test); *: significantly different compared to NC after 24 h of fermentation for neurotypical children (Kruskall–Wallis test with pairwise comparisons). Quantitative PCR (qPCR); log_10_ copies of 16S rRNA gene mL^−1^ of sample.

**Table 5 microorganisms-11-00414-t005:** Fecal microbial quantification after 24 h of fermentation in ASD children.

	NC	INU	scFOS	PO	PE	Overall *p*
Total bacteria	10.14 (10.05–10.18) ^†^	10.34 (10.33–10.37) ^a^	10.42 (10.33–10.48) ^a^	10.39 (10.31–10.54)	10.66 (10.42–10.75) *	0.005
*Bifidobacterium* spp.	8.83 (7.50–9.28) ^a^	10.11 (8.21–10.25) ^a^	10.15 (8.22–9.26) ^a^	9.72 (7.89–9.93) ^a^	9.56 (7.95–10.20) ^a^	0.132
*Lactobacillus* group	5.11 (4.75–5.71)	5.46 (5.05–5.64)	5.45 (5.06–5.56)	5.38 (5.11–5.57)	5.58 (5.28–5.76)	0.832
*C*. *perfringens* group	6.65 (5.45–7.21)	6.82 (5.53–7.66)	6.94 (5.53–7.73)	6.97 (5.71–7.60)	7.30 (5.86–8.54)	0.715
*Bacteroides* spp.	9.32 (9.14–9.48)	9.74 (9.53–9.89) ^a^	9.74 (9.47-9.89) ^a^	9.82 (9.50–9.86)	10.04 (9.75–10.07) *	0.029
*F*. *prausnitzii*	8.44 (8.40–8.70)	9.03 (8.80–9.20) ^†^	8.92 (8.82–9.16) ^†^	9.02 (8.82–9.35)	9.23 (8.92–9.60) *	0.045
*Prevotella* spp.	7.04 (6.59–7.53)	7.16 (6.93–8.33) ^a^	7.24 (7.05–8.47) ^a^	7.08 (7.00–8.33) ^a^	7.32 (7.15–8.63) ^a^	0.450
*A*. *muciniphila*	8.59 (6.32–8.99)	7.02 (5.55–8.97)	6.74 (5.63–8.97)	6.93 (5.80–8.97)	7.20 (5.88–9.00)	0.979

Values are expressed as median and Q1–Q3 quartiles for non-parametric data; ^a^: significantly different compared to t = 0 h for ASD children (Wilcoxon signed-rank test); *: significantly different compared to NC after 24 h of fermentation for ASD children (Kruskall–Wallis test with pairwise comparisons); ^†^: significantly different compared to neurotypical children in the case of each treatment (NC, INU, scFOS, PO, and PE) after 24 h fermentation (Mann–Whitney test). Quantitative PCR (qPCR); log_10_ copies of 16S rRNA gene mL^−1^ of sample.

**Table 6 microorganisms-11-00414-t006:** Total short-chain fatty acids (TSCFAs) and SCFAs concentrations (μmol mL^−1^ of sample), after 8 and 24 h of fermentation for the neurotypical children group.

Concentration μmol/mL												
	NC		INU		scFOS		PO		PE		Overall *p* Value	
	8 h	24 h	8 h	24 h	8 h	24 h	8 h	24 h	8 h	24 h	8 h	24 h
Total VFAs	11.88 (7.24–14.31) ^a^	22.11 (20.39–24.34) ^a,b^	35.62 (19.25–46.16) ^a^	56.00 (51.03–82.07) ^a,b^	34.76 (21.03–48.66) ^a^	73.51 (71.48–79.49) ^a,b^	33.79 (22.03–46.65) ^a^	71.03 (57.36–74.97) ^a,b^	39.78 (22.24–62.09) *^,a^	81.38 (76.27–84.40) *^,a,b^	0.017	0.005
AA	6.86 (4.54–8.70) ^a^	13.06 (11.46–13.61) ^a,b^	22.86 (14.13–29.03) ^a^	42.41 (28.33–63.88) ^a,b^	25.57 (15.70–35.04) *^,a^	67.79 (52.66–68.99) *^,a,b^	23.28 (16.41–27.21) ^a^	38.90 (33.21–39.80) ^a,b^	25.46 (16.73–32.49) *^,a^	44.82 (36.25–49.76) ^a,b^	0.018	0.002
PPA	1.92 (1.28–3.11) ^a^	2.77 (2.47–5.05) ^a,b^	4.24 (1.65–15.09) ^a^	7.04 (5.62–19.73) ^a,b^	3.93 (1.76–11.68) ^a^	5.18 (3.85–14.23) ^a,b^	4.69 (2.28–12.45) ^a^	11.19 (8.76–19.98) *^,a,b^	4.94 (2.31–22.79) ^a^	16.53 (9.92–27.63) *^,a,b^	0.327	0.010
BA	1.14 (0.64–1.97)	2.70 (1.95–3.59) ^a,b^	3.83 (2.05-4.61)	4.97 (3.95–10.85) ^a,b^	3.10 (1.20–4.38) ^a^	3.58 (1.52–7.11) ^a^	4.79 (2.34–6.98) ^a^	14.15 (10.39–17.52) ^a b^	4.78 (2.55–8.44) ^a^	16.45 (14.34–21.18) *^,†,a,b^	0.109	0.001
BSCFAs	0.46 (0.29–0.71) ^a^	2.39 (2.11–2.50)^a,b^	0.13 (0.10–0.31)	0.21 (0.16–0.41) ^b^	0.11 (0.09–0.27)	0.13 (0.09–0.30) *	0.25 (0.15–0.35)	0.38 (0.29–2.34) ^a,b^	0.19 (0.16–0.31)	0.28 (0.21–0.92) ^a,b^	0.056	0.006
Other	0.47 (0.24–0.57) ^a^	1.17 (0.74–1.25) ^a,b^	0.29 (0.20–0.38) ^a^	0.33 (0.24–0.46) ^a^	0.21 (0.17–0.31)	0.24 (0.19–0.33) *^,b^	0.29 (0.22–0.51) ^a^	0.82 (0.55–2.29) ^†,a,b^	0.27 (0.23–0.40) ^a^	0.60 (0.48–1.33) ^a,b^	0.297	0.002

Values are expressed as median and Q1–Q3 quartiles for non-parametric data; ^a^: significantly different compared to t = 0 h for neurotypical children and treatment (Wilcoxon signed-rank test); *: significantly different compared to NC after 24 h of fermentation for neurotypical children (Kruskall–Wallis test with pairwise comparisons); ^†^: significantly different compared to scFOS after fermentation for neurotypical children (Kruskall–Wallis test with pairwise comparisons); ^b^: significantly different compared to 8 h of fermentation for each condition and treatment (Wilcoxon signed rank test). AA: acetate, PPA: Propionate, BA: butyrate, BSCFAs: Branched short-chain fatty acids.

**Table 7 microorganisms-11-00414-t007:** Total short-chain fatty acids (TSCFAs) and SCFAs concentrations (μmol mL^−1^ of sample) after 8 and 24 h of fermentation for the ASD children group.

Concentration (μmol/mL)												
	NC		INU		scFOS		PO		PE		Overall *p* Value	
	8 h	24 h	8 h	24 h	8 h	24 h	8 h	24 h	8 h	24 h	8 h	24 h
TSCFAs	11.67 (11.44–18.27) ^a^	23.44 (22.94–28.34) ^a,b^	28.21 (22.80–41.49) ^a^	47.78 (39.33–66.32) ^a,b^	31.11 (26.93–47.15) ^a^	63.78 (46.38–79.41) ^a,b^	36.74 (30.63–44.75) ^a^	60.81 (51.83–69.15) ^a,b^	47.21 (36.06–52.26) *^,a^	78.61 (60.99–84.45) *^,a,b^	0.007	0.005
AA	8.03 (6.45–11.69) ^a^	14.58 (12.07–17.25) ^a,b^	17.74 (15.76–31.10) ^a^	33.82 (23.50–52.40) ^a,b^	22.96 (18.35–38.53) ^a^	52.91 (34.03–70.19) *^,a,b^	26.35 (19.85–27.67) ^a^	30.12 (27.82–37.95) ^a,b^	26.58 (23.52–37.06) *^,a^	40.53 (30.59–52.39) ^a^	0.013	0.012
PPA	2.21 (1.74–2.63) ^a^	3.80 (3.18–4.12) ^a,b^	4.37 (2.28–4.65) ^a^	5.80 (3.20–6.22) ^a,b^	3.51 (2.03–5.24) ^a^	3.87 (2.39–6.24) ^a,b^	6.94 (3.64–7.40) ^a^	9.69 (9.29–12.13) *^,a,b^	7.88 (3.33–11.22) *^,a^	12.63 (9.79–17.90) *^,†,a, b^	0.029	0.001
BA	1.80 (1.62–2.56)	3.34 (2.81–4.91) ^a,b^	5.04 (3.09–6.68) ^a^	9.35 (5.61–11.71) ^a,b^	4.99 (2.19–6.50) ^a^	6.10 (2.54–9.75) ^a^	7.39 (3.29–10.28) ^a^	17.86 (10.18–20.37) *^,a,b^	6.99 (2.73–11.61) ^a^	19.97 (7.35–26.23) *^,a,b^	0.042	0.010
BSCFAs	0.53 (0.37–1.27) ^a^	2.25 (2.24–2.30) ^a,b^	0.22 (0.19–0.34) ^a^	0.48 (0.43–0.92) ^a,b^	0.20 (0.15–0.31) *^,a^	0.24 (0.18–0.28) *	0.25 (0.21–0.35) ^a^	0.59 (0.31–2.32) ^a,b^	0.29 (0.24–0.63) ^a^	0.91 (0.29–1.55) ^a,b^	0.030	0.003
Other	0.32 (0.24–0.54) ^a^	0.84 (0.78–0.94) ^a,b^	0.23 (0.18–0.32)	0.28 (0.26–0.45) ^a,b^	0.17 (0.15–0.27)	0.19 (0.16–0.29) *	0.17 (0.15–0.37)	0.32 (0.24–1.48) ^b^	0.15 (0.13–0.51)	0.37 (0.20–1.25) ^a,b^	0.257	0.029
Other	2.72 (1.79–3.52)	3.41 (3.00–3.89)	0.71 (0.63–1.04) ^a^	0.62 (0.41–1.01) ^a^	0.54 (0.41–0.82) *^,a^	0.30 (0.22–0.58) *^,a^	0.55 (0.47–0.86) ^a^	0.53 (0.46–2.22) ^a^	0.39 (0.32–1.05) *^,a^	0.51 (0.34–1.49) ^a^	0.007	0.022

Values are expressed as median and Q1–Q3 quartiles for non-parametric data; ^a^: significantly different compared to t = 0 h for ASD children and treatment (Wilcoxon signed-rank test); *: significantly different compared to NC after 24 h of fermentation for ASD children (Kruskall–Wallis test with pairwise comparisons); ^†^: significantly different compared to scFOS after fermentation for ASD children (Kruskall–Wallis test with paiwise comparisons); ^b^: significantly different compared to 8 h of fermentation for each condition and treatment (Wilcoxon signed rank test). AA: acetate, PPA: Propionate, BA: butyrate, BSCFAs: Branched short-chain fatty acids.

## Data Availability

The data presented in this study are available upon request from the corresponding author.

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
