# Peer review of "In Vitro Fermentation of Edible Mushrooms: Effects on Faecal Microbiota Characteristics of Autistic and Neurotypical Children"

_microorganisms, 2023, doi:10.3390/microorganisms11020414_

Round 1

Reviewer 1 Report

The reviewed manuscript touches on the very interesting and often recently discussed problem of the influence of the gut microbiota on the nervous system. And especially on the positive effects of beneficial microbiota on various types of dysfunctions of the nervous system. In this case, the study investigated the effect of lyophilized fruiting bodies of Pleurotus eryngii and P. ostreatus on changing the microbiota of feces collected from children with autism spectrum disorder (ASD). The research conducted by the authors was planned and carried out in an exemplary manner. All chapters are adequately and comprehensively described. The hypothesis established by the authors has been confirmed and properly explained. The authors showed that the fruiting bodies of P. eryngii and P. ostreatus can represent a potentially novel prebiotics with possible beneficial effects on ASD children. Particularly noteworthy is the demonstration of the bifidogenic effect of the lyophilizates used. Overall, the entire reviewed work should be considered of great value. However, after reading the manuscript, I have some doubts about the title. Yes, Pleurotus fruiting bodies are known for their high content of β-glucans, but they contain a large amount of other polysaccharides, such as α-(1,3)-glucans and mannans. Therefore, without using β-glucans as a positive control in the study, we cannot talk about the effect of these polymers. For the proposed title to be appropriate, the authors should use Pleuran as a control in their study. And since there was no such control I suggest changing the title. In addition to this main point, I have some minor comments. In Section 3.2, the notation of the names of microorganisms should be changed to italics.

Reviewer 2 Report

The manuscript by Georgia Saxami et al. investigated the effect of edible mushrooms rich in β-glucans on faecal microbiota characteristics from autistic and neurotypical children. The study is of interest to the readers and I have the following comments and suggestions:

1, The authors must analyze the β-glucans contents in the mushrooms. This is very important for the present study. We must know how much β-glucans were added to the medium. 

2, Table 1, 2 and 4 should be three-line tables. The authors must revise. 

3, The statistics should be added to Figure 1. Are the results statistically significant?

4, Why only 5 samples were used in each group? This is a limitation of the present study. More likely, this is a proof-of-concept study. This must be discussed in the manuscript. 

5, Why did not the authors use 16S sequencing to check the changes of the gut microbiota? This would give more information. 

Reviewer 3 Report

This manuscript explored that beta-glucan-rich mushroom can be a probiotic for autistic and neurotypical children. As this paper is not written in a concise manner, I checked 'Extensive editing of English language and style required'. The following is a list of comments.

1-The results of the baseline (Table 4 and 5) and their explanation in the text can be deleted.

2-The problem is that changing the units changes the results of the statistics. Molar ratio in Tables 8 and 9 can be deleted.

3-The authors did not analyze all gut bacteria but only seen bacteria. They did not explain the reason in the text.

4-The total VFAs should change to total SCFA.

5-The authors investigated autistic and neurotypical children. They need to explain why they investigated neurotypical children but not normal children for readers.

Round 2

Reviewer 2 Report

The authors have revised the manuscript. It can be considered for publication. 

Author Response

We thank the Reviewer for his/her efforts

Reviewer 3 Report

The authors have not properly responded to this reviewer's comments and revised the paper.

1-The previous reviewer’s commented is ‘The results of the baseline (Table 4 and 5) and their explanation in the text can be deleted’.

Although I understand authors’ response, Tables 4 and 5 should be written as in Table 6 because adding samples such as mushroom does not change the gut bacteria. In this paper, the authors should focus on two points; what is different between the gut bacteria of ASD and nurotypical children, and what did they find among each test after adding samples and fermenting? The explanation and discussion including "baseline" must confuse the readers. Therefore, they should delete all descriptions of "baseline".

2-The previous reviewer’s commented is ‘The problem is that changing the units changes the results of the statistics. Molar ratio in Tables 8 and 9 can be deleted’.

In general, tables should not be described in different units in terms of conciseness. In addition, this manuscript does not discuss the need to do so in the text.

3-The previous reviewer’s commented is ‘The authors did not analyze all gut bacteria but only seven bacteria. They did not explain the reason in the text.

Again, they need to explain why they choose only seven bacteria in the text. This reviewer did not ask Lines:737-741.
